# Healthy Taiwanese Eating Approach (TEA) toward Total Wellbeing and Healthy Longevity

**DOI:** 10.3390/nu14132774

**Published:** 2022-07-05

**Authors:** Wen-Harn Pan, Szu-Yun Wu, Nai-Hua Yeh, Shu-Yi Hung

**Affiliations:** 1Institute of Biomedical Sciences, Academia Sinica, 128 Sec. 2, Academia Rd. Nankang, Taipei 115, Taiwan; s.wu@ibms.sinica.edu.tw (S.-Y.W.); naihua@ibms.sinica.edu.tw (N.-H.Y.); shuyihung@ibms.sinica.edu.tw (S.-Y.H.); 2Institute of Population Health Sciences, National Health Research Institute, 35, Keyan Road, Miaoli County 350, Taiwan; 3Department of Biochemical Science and Technology, National Taiwan University, No. 1, Sec. 4, Roosevelt Rd., Taipei 10617, Taiwan

**Keywords:** reduced rank analysis, partial least square discriminant analysis, dietary pattern, planet health, lactose intolerance, cooking method

## Abstract

A healthy dietary pattern review for Asian countries is scarce, which is crucial for guiding healthy eating. We reviewed Taiwanese dietary pattern discovery studies. Included were 19 studies, the majority of which employed dimension reduction methods to find dietary patterns associated with various health conditions. To show what is a high or low intake of foods in Taiwan, we also report the average dietary content and the 25th and 75th percentile values of the adult population for six food groups gathered by the Nutrition and Health Survey in Taiwan, 2017–2020. The healthy Taiwanese dietary approach is cohesive across multiple health outcomes occurring at different ages. It is featured with higher intakes of plant-based foods, aquatic foods, and some beneficial ethnic foods (soy products), drinks (tea), and cooking methods (boiling and steaming); lower intakes of fast foods, fatty and processed meats, sugar, salt rich foods/drinks, and fried foods; but with mixed findings for dairy and egg. Yet, the average Taiwanese person consumed many refined staple foods and livestock, but not sufficient vegetables, fruits, whole grains and roots, beans, and nuts. Dairy consumption remains low. In conclusion, Taiwanese discovery studies point to a mortality-lowering total wellbeing dietary pattern consistent with the current knowledge, which discloses potential benefits of soy product, tea, and boiling and steaming.

## 1. Introduction

What people eat or drink has profound impacts on health and total wellbeing. A prudent dietary pattern should be something suitable for people to follow throughout their whole lifespan. The Mediterranean diet and the DASH diet have been advocated as the currently best available dietary patterns for optimizing health [1]. However, Asian dietary patterns differ from the Western ones not only in food choices of major food groups, but also in cooking methods and drinks.

In Taiwan, the major staple foods are steamed white rice, white flour-made noodles, and dumpling skins. For lunch and dinner, pork, chicken, sea foods, tofu, and egg in that order, either stirred-fried, pan-fried, fried, or minced, are major sources of protein foods. Vegetables are mainly stirred fried with non-tropical plant oil. Tea is a major traditional drink with or without a meal, although a large number of various kinds of sugar-containing beverages are competing with tea and water. For breakfast, either traditional rice porridge, quick oatmeal with non-fat milk powder, high fat-containing sweet Taiwanese pastry with soymilk, or sandwiches with milk tea are all very popular. Compared to the US or other Western countries, the current mean servings of dairies, beef, and animal fat are much lower in Taiwan; but those of rice, pork, soy products, vegetables, and fruits are much higher [2]. 

It is timely to examine whether healthy Taiwanese dietary patterns have similar or unique features with those of DASH and Mediterranean ones. No matter what similarities or differences are discovered, they may provide insight into components of diet that work toward better or worse health. In this endeavor, we reviewed data mining-related literature [3,4,5,6,7,8,9,10,11,12,13,14,15,16,17,18,19,20,21] pertaining to Taiwanese dietary patterns and multiple health or disease outcomes and also described the recent Taiwanese dietary content using data collected in the Nutrition and Health Survey in Taiwan from 2017 to 2020 in order to shed light on healthy eating patterns and options in Asians. 

## 2. Materials and Methods

### 2.1. Strategy to Search Taiwanese Dietary Pattern-Related Literature

Medline and Cochrane library databases were searched up to April of 2022. Selected were studies in English and in full text. First, we searched for papers with the following keywords: “Taiwan∗” and “dietary pattern∗”. A total of 133 papers were found. Among them, 84 were not relevant. Then, excluded were another 12 studies that studied the effects of known dietary patterns such as the Mediterranean diet, vegetarian diet, Western diet, and healthy diet arbitrarily defined by authors. One article was left out due to a substantial amount of missing values. In addition, the results from complete data and those from imputed data were very different with respect to directions of the associations between dietary items and biomarkers of health outcomes.

The resulting 36 articles are composed of 17 studies employing “reduced rank regression (RRR) methods”, 1 study using “partial least square discriminant analysis (PLSDA)”, 11 studies applying principal component analysis (PCA), 6 studies utilizing the exploratory factor analysis (EFA), and one that univariately selected significant associated food items and combined them into a food score. In order to include only dietary factors highly associated with disease or health conditions of interest, we excluded dietary factors discovered by PCA or EFA that focused purely on dimension reduction of the multivariate dietary frequency variables. A total number of 19 articles were included in this review [3,4,5,6,7,8,9,10,11,12,13,14,15,16,17,18,19,20,21]. The flow chart of searching literatures is shown in Figure 1. 

### 2.2. Data Summary and Presentation Method 

For the ease to eyeball the dietary patterns, studies with similar ages are arranged together from the young (top) to the old (bottom) and the results of all 19 studies are summarized such that protective foods are colored green and risk foods are colored orange. We then used a simple approach to summarize the findings from the 19 studies, counting the number of times that a given food category was extracted into protective dietary factors and risky dietary factors, respectively. If gender groups are analyzed separately, a food category may be counted twice. A food category is considered protective and colored green, if it was at all times contained in an inversely associated (with disease outcomes) dietary factor. A food group with a negative loading coefficient within a positive-associated dietary factor was also consider protective and colored green. On the other hand, a category considered risky and colored orange, if it was at all times within a positively associated dietary factor and so were those with negative loading coefficients but within an inversely associated factor. For food categories in between the above two extremes, they were colored yellow. 

The protective foods are positioned on the left and the risky foods are on the right of the Table 1. Since the number of studies is small for cooking methods and eating places, the findings on protective methods, eating at home, and eating out are presented in the far-right. They were colored with a different shade of green (protective), and a different shade of orange (risky). The loading coefficients of all food categories beyond the selection threshold determined by authors are also listed in the table. No loading coefficients were provided for one article (20), which found significantly associated food items by univariate logistic regression method.

### 2.3. Estimating Taiwanese Current Dietary Content

Data on 24-h recall from the Nutrition and Health Survey from Taiwan 2017–2020 was used. Design and operation of this survey has been consistent with previous surveys that have been described elsewhere [22]. Included were those who were aged 19 and above at the time of data collection. Data from a total of 6538 participants (3235 males and 3303 females) were included in the analysis. Mean intake value and those at the 25th, 50th, and 75th percentiles were estimated for “grains and roots”, “dairies”, other “protein-rich foods”, “fats/oils/nuts”, “vegetables”, and “fruits”. Food frequency information was used to estimate the mean consumption frequency of coffee and tea and those at the 25th percentile, 50th, and 75th percentile points. Sample weights have been incorporated in the estimation process to provide representative statistics with SAS version 9.4.

## 3. Results

### 3.1. Studied Populations and Design Characteristics of the Taiwanese Dietary Pattern Data Mining Studies

From the 19 manuscripts, we found 2 prospective, 16 cross-sectional, and 1 case-control studies (Table 1). One prospective study employed data from the Mei Jau Health Institute check-up cohort (*n* = 62,645) [16], and the other one from the Nutrition and Health Survey in Taiwan (*n* = 2475) [20]. A total of five cross-sectional studies (*n* = 25,569–118,924) used data from the MJ health check-up program [8,9,10,11,17], five studies (*n* = 1245–3071) utilized the data from the Nutrition and Health Survey in Taiwan [4,5,6,12,15], four studies (*n* = 125–212) derived data from the Taipei Medical University Hospital study [3,7,13,21], one study (*n*= 2397) applied data from the Taiwan Children Health Study [14], and another one study employed data from the Taiwan Longitudinal Survey of Aging (TLSA) study (*n* = 3486) [18], respectively. In addition, there was one case-control study on nasopharyngeal carcinoma (*n* = 372 vs. 378) by the National Taiwan University and MacKay Memorial Hospitals [19]. The age ranges covered by these Taiwanese dietary pattern studies are wide, including children aged 5 years to 12 years, young and middle-aged adults, and elderlies.

### 3.2. Health Outcomes of Interest of the Taiwanese Dietary Pattern Data Mining Studies

For children, dietary patterns for asthma and/or other allergic diseases were studied (Table 1) [12,14]. For those at early or middle age adulthood, explored were a wide range of health conditions, including anemia [10,17], obesity/central obesity [3,4], dyslipidemia [10,13], metabolic syndrome [3,9,13,21], hyperuricemia [5], blood testosterone level [7,8], liver function [16], kidney function [11], and all-cause and cause-specific mortality [20]. For elderlies, dietary patterns for frailty and cognitive declines have been studied [6,15,18]. In addition, nasopharyngeal carcinoma cases diagnosed below age 75 were compared with aged and neighborhood-matched community-based controls [19].

Food categories that have been included in positively or inversely associated dietary factors in the 19 literatures are summarized in Table 2.

### 3.3. Food Items Identified as Inversely Associated with Biomarkers or Disease Outcomes

Within constructed dietary factors across 19 studies, food categories that were exclusively and inversely associated with health outcomes including all-cause and CVD mortalities are vegetables (15 times) [5,6,7,8,10,11,12,13,15,16,18,19,20,21], fruits (12 times) [4,6,9,10,11,12,15,16,18,19,20], seafood (11 times) [5,6,10,13,15,16,18,19,20,21], beans/soybean products (6 times) [5,6,8,11,21], tea (6 times) [4,6,15,18,19,20], whole grains (5 times) [4,6,10,15], and nuts (4 times) [4,6,15] in that order, pointing to the beneficial impact of consuming plant foods (vegetables, fruits, beans, whole grains) and drinks (tea) as well as aquatic foods under the current food consumption profile of Taiwan. Coffee is a plant-based drink and has been included in the protective dietary factors of three studies on hyperuricemia, morbid obesity, and mild cognitive impairment, respectively [4,5,6]; but in one study [3], coffee consumption frequency was positively associated with central obesity and metabolic syndrome. In this study, coffee was not separated to subcategories of plain, sugar-containing, or sugar- and cream-containing (personal communication with author) coffee and it is also a part of a dietary pattern characterized by high frequencies of deep-fried foods, processed meats, chicken, pork, eating out, and animal fat/skin, but low intake frequencies of steamed/boiled/raw foods and dairy products [3].

### 3.4. Food Items Identified as Positively Associated with Biomarkers or Disease Outcomes

Within constructed dietary factors across 19 studies, food categories that exclusively and positively associated disease outcomes are processed foods (10 times) [3,4,5,8,9,10,11,16,17,20], internal organs or marble meats (8 times) [3,5,8,9,10,11,17,20], sugar-containing beverages and candies (7 times) [4,5,10,12,16,17,20], fast foods (6 times) [3,8,9,11,12,13], dipping sauces (4 times) [8,9,11,16], refined desserts (3 times) [7,12,21], bamboo shoots (in one study for hyperuricemia [5]), and jam or honey (once) [10], indicating that disease risks were associated with consuming more of the processed foods, sugary drinks, sauces rich with sugar and salt, and high saturated fatty acid-containing animal foods. Since bamboo shoot is a purine-rich vegetable, it is not surprised to be linked with hyperuricemia. Meat consumption frequency has been included in risk elevating dietary factors 7 times among the 19 studies [3,4,9,10,11,14,17]. Nonetheless, one study showed that lean meat is inversely associated with hyperuricemia in contrast to the positive association with organ meat in that study [5]. In addition, meat was inversely associated with frailty in older adults [18].

### 3.5. Food Items Bidirectionally Associated with Biomarkers or Disease Outcomes

Steamed rice and boiled noodles, two popular staple foods in Taiwan, have been included in dietary factors inversely associated with asthma [12], hypogonadism [7], and frailty [18]; however, they were positively associated with metabolic syndrome [9,21]. One study showed inverse association with dyslipidemia/metabolic syndrome for steamed rice-related factors, but positive association for boiled noodle-related factors [13]. On the contrary, fried rice or noodle products were all within positively associated dietary factors for poor testicular function [8], anemia [17], obesity/CVD risk/impaired kidney function [11], and metabolic syndrome [9].

Although dairy consumption frequency was within dietary factors inversely associated with the following disease outcomes: nasopharyngeal carcinoma [19], poor testicular function [8], obesity [3,4], metabolic syndrome [13,21], frailty [15], mild cognitive decline [6], and all-cause and CVD mortality [20], in a study of elementary school students [12], it was positively associated with allergic disease such as asthma within a factor containing high intakes of dessert, fast food, and sugar-containing beverages and low intakes of vegetables and fruits. In another study [7], dairy consumption frequency was also positively associated with low serum testosterone values (hypogonadism) imbedded in a dietary factor with a higher frequency of consuming pastry, dessert, and eating out, but a lower frequency of consuming vegetables and cooking at home or cooking with boiling or steaming.

Egg consumption frequency has been inversely associated with nasopharyngeal carcinoma [19], abnormal liver function [16], hyperuricemia in women [5], frailty in older adults [18], and mild cognitive impairment in elderly women [4] in four separate studies; but positively associated with anemia [17], dyslipidemia [10], and metabolic syndrome [9] in another three studies.

Three studies have identified consumption frequency of Taiwanese style pastries in disease-associated dietary factors [7,9,11]. One study showed a positive association with hypogonadism [7]. Two studies demonstrated an inverse association with obesity/CVD/ impaired kidney function and metabolic syndrome, respectively [9,11].

### 3.6. Cooking Methods, Eating Out, and Disease Outcomes

There were just four studies that identified an association with cooking methods or cooking at home/eating out [3,7,13,21]. The frequency of steaming and boiling foods has been inversely associated with dyslipidemia/metabolic syndrome/central obesity [3,13,21]. Cooking at home has been inversely associated with hypogonadism in one study [7], but positively with metabolic syndrome in another study [21]. On the contrary, eating out frequency was all positively associated with disease outcomes including central obesity/metabolic syndrome and hypogonadism [3,7].

### 3.7. Recent Dietary Content of Taiwanese Adults 

As showing in Table 3, Taiwanese recently eat on average 12.5 servings (15 g carbohydrate per serving) of carbohydrate-rich foods a day. Total carbohydrate (including added sugar) equates roughly to that from 3.1 bowls of rice a day including added sugar. Among these, around 24% were from refined convenience foods, sweet pastries, cakes, and cookies and 13.6% were from either sugar added in cooking or contained in beverages. A mean of 7.7 servings (7 g protein per serving) of protein-rich foods is consumed daily, in which livestock (mainly pork) and poultry constitute 4.1 servings, i.e., more than half of all protein foods. In addition, a little over 1 serving (1.3 per day) of fish/seafood and 1.1 serving of soy products and 0.8 servings of egg were consumed daily. In terms of dairies, only half a serving was eaten a day. In addition, around 2.4 servings of vegetables, 1.5 servings of fruits, 0.5 servings of seeds/nuts, and 4.8 servings of cooking oils were consumed daily, respectively. With respect to natural drinks, Taiwanese on average drink 0.43 and 0.26 times of sugar-free tea and coffee a day, respectively. Among all the above categories, the consumption weight or frequency distributions of pastry/cookies, seeds/nuts, dairies, and black coffee are all extremely skewed to the right, with a median of zero.

With respect to percentile values, the 75th percentile values of vegetables, fruits, and nuts, are 3.3, 2.2, and 0, respectively. The 75th percentile value for grains and roots is 10.3 servings (roughly 2.5 bowls of steamed rice) and most of them are in refined forms. For protein-rich foods, the 25th, 50th, and 75th percentile values of livestock and poultry, aquatic foods, soybean products, and egg are 1.2, 3.1, 5.7; 0, 0.4, 1.8; 0, 0.1, 1.5; and 0, 0.5, 1.1 servings, respectively. The 25th and 50th percentile values for plain tea and coffee consumption frequency are all near zero, indicating that the majority of Taiwanese do not drink coffee or tea. But the 75th percentile values are 0.71 and 0.29, respectively, which added together equates to one cup of caffeine-containing beverage a day.

## 4. Discussion

In this study, we propose an evidence-based, healthy Taiwanese eating approach (TEA) given the current eating culture and daily practice. We reviewed 19 Taiwanese dietary pattern discovery studies published up to April of 2022, most of which employed dimension reduction methods to reduce information for food frequency data, while at the same time maximizing variation of the studied outcomes explained by the food frequency matrix. Information included in this review covers a wide range of health conditions such as allergic diseases of the young, various cardio-metabolic diseases in middle aged adults (such as obesity, metabolic syndrome, dyslipidemia, hyperuricemia, anemia, and poor testicular and liver functions), as well as cancer, frailty, and cognitive decline in older age. In addition, one study used all-cause mortality as the major outcome variable. Despite different diseases or conditions at different ages, we can find consistently beneficial and deleterious dietary components of health outcomes including those on healthy longevity, indicating there is indeed a prudent dietary pattern for total wellbeing with which people can practice throughout the whole lifespan and just make subtle modifications depending on gender, life stages, and individualized disease profiles.

We found that the healthy TEA is one featured with higher intakes of plant-based foods (vegetables, fruits, soy protein foods, whole grains, and nuts) and plant-based drinks (tea) and aquatic foods, but limited in processed foods, internal organs, fatty meats, sugar-containing beverages or candies, fast foods, dipping sauces, fried rice or noodle products, and refined desserts. This pattern is consistent with the DASH and Mediterranean diets, but with its unique culture-specific features such as more soy products and tea, but less fried rice or noodle products and internal organs/marbled pork. This dietary pattern has been associated with lower all-cause and CVD-mortalities in one of the articles [20]. 

The beneficial effects of vegetables and fruits [23], whole grains [24,25], nuts [26], and omega-3-rich fish [27] have been well-documented. The unique features found in this review are those of soy products and tea. Soy consumption has been linked to many health benefits in reducing non-communicable diseases or conditions such as immune disorders, obesity, cardiovascular disease, metabolic syndrome, and certain types of cancer [28]. Among bioactive substances, polyunsaturated fatty acids are critical in lowering blood cholesterol and many soy peptides have hypolipidemic, antihypertensive, anti-inflammatory, and immunomodulatory properties. The Taiwanese average consumption frequency for soy products is around once a day, which is much less than that of livestock and poultry. Due to the versatile ways of preparing and enjoying soy products in Taiwan and in other Asian countries and that the top 25 percent of the Taiwanese consume more than 1.5 servings a day, we feel that it is not difficult to push to a level of two servings (types) of soy products a day: for example, one glass of soy milk for breakfast and one serving of tofu (hard or soft) or fresh soybean at lunch or dinner, prepared in miscellaneous ways. 

Tea is a popular and socially welcomed plant-based drink in Taiwan due to a long history of tea drinking. Tea is made into various flavors, green or fermented. A growing body of evidence suggests that frequent tea consumption may protect against noncommunicable diseases including the formation of kidney stones, bacterial infections, dental caries, cardiovascular disease, and various types of cancers [29]. The Taiwanese average consumption frequency for tea is around 2–3 times a week. People at the 75th percentile is around 5 times a week. Since almost all Taiwanese families keep some types of tea at home and it is often served at home or restaurants, it should not be a problem to push to a Taiwanese higher percentile value such as once a day. Since the 75th percentile value for coffee is about two times a week in Taiwan. A mixture of plain tea and coffee consumption pattern is also acceptable. 

Since dairies concurred with an increased risk of allergy in some Taiwanese studies and controversy still exists for its consumption [30], they may be consumed with moderation or avoided by people with a dairy-associated allergy or with familial or genetic propensity of this condition. Due to the high prevalence of lactose intolerance [31], milk consumption has remained low at a mean level of 0.5 glasses a day and limited to a small proportion of people (milk intake median is 0) despite the governmental promotion policy in past decades. The current recommendation of 1 to 2 glasses of milk a day seems unrealistic for the Taiwanese ethnicity and does not comply with the current trend of reducing beef and dairy cow husbandry for planet health [32]. Since the Taiwanese food guide recommended 1–2 glasses of milk a day for achieving RDAs of calcium, magnesium, and vitamin B2 and since vitamin D, vitamin B2, and calcium insufficiencies remain as problems in Taiwan, we would suggest fortifying the popular soy milk or other plant-based milk with calcium, vitamin D, and vitamin B2, etc., in order to meet both the nutritional and environmental needs.

According to our descriptive data, the Taiwanese currently consume rice, pork/chicken, and non-tropical plant oils as the major staples, and protein food, cooking oil, moderate amounts of vegetable/fruit, fish/seafood, tofu, along with relatively low level of dairies and nuts. Among the staple foods and protein-rich foods, processed foods such as fried rice and noodle products, pastry/cookies, sugar-containing beverages/candies, and processed meat/seafood constitute a nontrivial proportion. This consumption pattern is nowhere near the Taiwanese daily food guide (Table 3). On average, vegetables, fruits, dairies, and nuts all fall short of the recommendation. The pork (a major red meat) and poultry category at four servings a day leads the protein-rich foods, while just roughly one serving of soy product, seafood, and egg are consumed daily, respectively. Under Taiwanese food culture, people are customarily to drink more tea and soy milk and consume more tofu, peanuts, and sesame than Western counterparts. In addition, the aquatic food supply is abundant from the island surroundings and aquaculture industry. We shall call for increasing the proportion of whole foods, reducing processed ultra-refined foods, and consuming more soy products and aquatic foods to replace red meat for the benefits of both human and planet health. We would suggest revising the Taiwanese food guide toward a plant-based one as shown in Table 3, taking the 75th value for beneficial protein foods (soy products and aquatic foods) and the 25th value for livestock and poultry, while maintaining the original, yet none-reaching goals of five servings of vegetables and fruits and one serving of nuts/seeds. 

Although meat was mostly associated with risk elevation in this review, it was protective of frailty in older age [33], indicating that protein-rich foods should be encouraged when overall food intake is low in older ages. In addition, egg consumption concurred with risk elevation for dyslipidemia and metabolic syndrome in some Taiwanese studies, while recent literature shows no association between egg consumption and cardiovascular diseases [34]. Although one egg a day seems a reasonable recommendation for general public, caution should be taken for those with dyslipidemia and cardiovascular diseases. 

Taiwan and Japan are both countries surrounded by sea. Food supplies are similar. However, a major difference in dietary content is due to cooking methods. The Taiwanese use stirred frying as the major cooking method and fried pork chop, chicken, and alike are very popular, while the Japanese enjoy raw fish and sushi. Therefore, the percent calories from fat are much higher in Taiwan compared to Japan. Although there were just three studies in this review investigating cooking methods and eating out/at home and disease outcomes, they all point to the benefits of steaming or boiling foods and preparing foods at home.

Finally, with respect to the limitation of the current study, we feel that our discovery is limited by the dimension of the dietary data. Since the food frequency questionnaire allows only a modest amount of questions and the human being has a limited ability to recall types and quantity of certain food groups, particularly those with many varieties such as fruits and vegetable, the effects of various phytonutrients or other unique substances in foods and spices cannot be uncovered. Modern technology of dietary data collection may be used to aid detailed dietary data collection that in turn may help to find more beneficial and harmful foods in the future.

## 5. Conclusions

We discovered that, across several big Taiwanese databases, healthy TEA summarized from data mining studies are consistent irrespective of the disease or health conditions under consideration. A healthy TEA is characterized by a greater consumption of plant-based foods (vegetables, fruits, whole grains, nuts, and seeds) and drinks (tea) as well as aquatic foods, but less marbled meat, processed foods, high-fat (or fried) and high-sugar snacks, and sugar-containing beverages. A healthy TEA is similar to the DASH and Mediterranean diets, but points to some beneficial ethnic food, drink, and cooking methods, such as drinking tea, consuming soybean products, and using more boiling and steaming methods for cooking. Thus, according to the above, we have made a recommendation for a new Taiwanese food guide and hope to arouse attention in Asian countries with a similar food culture.

## Figures and Tables

**Figure 1 nutrients-14-02774-f001:**
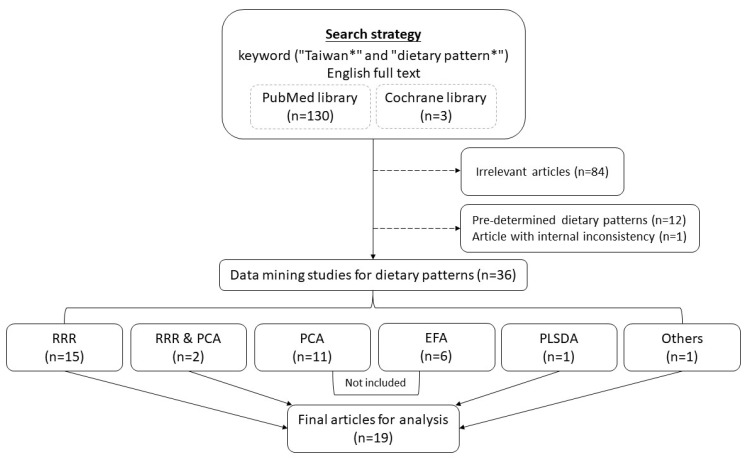
Flow chart of literature search. EFA, exploratory factor analysis; PCA, principal component analysis; PLSDA, partial least square discriminant analysis; RRR, reduced rank regression.

**Table 1 nutrients-14-02774-t001:** Description of the included data mining dietary pattern studies.

Ref.	Participants’ Characteristics	Outcome	Data Set	Study Design	Methods and Cut-Off Point
[12]	*n* = 2082, 7–12 y	Asthma	NAHSIT 2001–2002 (NAHSIT Children)	Cross-sectional	RRR, >0.24 and <−0.27
[14]	*n* = 2397 4th graders	Respiratory diseases	Taiwan Children Health Study 2011	Cross-sectional	RRR, 0.3
[19]	*n* = 372 controls and 378 cases <75 y with incident, primary, histologically confirmed NPC	NPC	The National Taiwan University and MacKay Memorial Hospitals patients 1991–1994	Case-control	PLSDA, 0.2
[8]	*n* = 3283, men	Poor testicular function	MJHID 2009–2015	Cross-sectional	RRR, 0.2
[20]	*n* = 2475, 18~65 y	All-cause and cause-specific mortality	NAHSIT 1993–1996	Prospective	Significant univariate predictive food item, *p* < 0.05
[17]	*n* = 118,924, 20–45 y	Anemia	MJHID 2001–2015	Cross-sectional	RRR, 0.2
[16]	62,645 participants met the criteria: (1) aged between 20 and 45 years, (2) free of chronic diseases, (3) having normal levels of serum liver enzymes at baseline and no history of receiving hepatic treatment, (4) having at least one follow-up visit, and (5) having complete data at baseline. 11,506 participants developed abnormal liver function (18.4%).	Abnormal Liver Function	MJHID 2001–2015	Longitudinal	RRR, 0.2
[13]	*n* = 212 adults, 20–64 y	Hyperlipidemia and Metabolic syndrome	TMUH 2015–2016 patients at the Division of Gastroenterology and Hepatobiliary Diseases, Department of Internal Medicine,	Cross-sectional	RRR, 0.2
[7]	*n* = 125, 20–64 y men	Hypogonadism	TMUH 2015 patients at the Division of Gastroenterology and Hepatobiliary Diseases, Department of Internal Medicine,	Cross-sectional	RRR, 0.2
[21]	*n* = 166, 20–64 y	Metabolic syndrome	TMUH 2015	Cross-sectional	RRR, 0.2
[3]	*n* = 208, 20–65 y	Central obesity and Metabolic syndrome	TMUH 2015–2016 patients at the Division of Gastroenterology and Hepatobiliary Diseases, Department of Internal Medicine,	Cross-sectional	RRR, 0.2
[5]	*n* = 2979 and 1661	Hyperuricemia	NAHSIT 1993–1996, 2005–2008	Cross-sectional	RRR, 0.2
[4]	*n* = 3071, 1673, 1440, all over 19 y	Morbid obesity	NAHSIT 1993–1996, 2005–2008, 2013–2014	Cross-sectional	RRR, 0.15
[10]	*n* = 41,128, eGFR <90 mL/min/1.73 m^2^ and positive urinary protein	Dyslipidemia in males and anemia	MJHID 2008–2010	Cross-sectional	RRR, 0.2
[11]	*n* = 41,128, eGFR <90 mL/min/1.73 m^2^ and proteinuria	Weight status, increased cardiovascular risk, and severity of impaired kidney function	MJHID 2008–2010	Cross-sectional	RRR, 0.2
[9]	*n* = 25,569 over 40 y, eGFR <90 mL/min/1.73 m^2^, and positive urinary protein	Metabolic syndrome	MJ 2008~2010	Cross-sectional	RRR, 0.2
[18]	*n* = 3486, 53 y and above	Frailty	TLSA (1999, 2003, 2007, and 2011)	Cross-sectional	RRR, 0.2
[15]	*n* = 1440, 65 y and above	Frailty	NAHSIT 2014–2016	Cross-sectional	RRR, 0.2
[6]	*n* = 1245, 65 y and above	Mild cognitive impairment	NAHSIT 2014–2016	Cross-sectional	RRR, 0.2

eGFR: estimated glomerular filtration rate, MJHID: Mei Jau Health Institute database, NAHSIT: Nutrition and Health Survey in Taiwan, NPC: Nasopharyngeal carcinoma, PLSDA: partial least square discriminant analysis, RRR: reduced rank regression, TLSA: Taiwan Longitudinal Study on Aging, TMUH: Taipei Medical University Hospital.

**Table 2 nutrients-14-02774-t002:** Food items identified as inversely, positively, and bidirectionally associated with disease-associated biomarkers and outcomes including mortality in 19 discovery dietary pattern studies **^a^**.

				Factor Loading for Each Food Item (Protective Counts, Risky Counts)																								
Ref	Age (y)	Outcome ^b^	Response Variables	Vegetable (15,0)	Fruit (12, 0)	Seafood, Fish, or Both (11,0)	Bean or Soy Bean Product (6,0)	Tea (6,0)	Whole Grains (5,0)	Nuts (4,0)	Coffee (5,1)	Rice (r) or Noodle (n) Not Fried (3,3)	Dairy (9,3)	Egg (5,3)	Bread & Pastry (2,1)	Jam or Honey (0,1)	Bamboo Shoots (0,2)	Refined Dessert (0,3)	Fried Rice or Noodle Product (0,4)	Sauce (0,4)	Fast Food (0,6)	Meat (2,7)	Sugary Drinks or Candies (0,7)	Internal Organs or Marble Meat (0,8)	Processed Products (0,10)	Healthy Cooking Methods (3,0)	Eating at Home (1,1)	Eating Out (0,2)
[12]	7–12	Asthma	Asthma symptom score	−0.44	−0.34							−0.27r	0.31					0.44			0.25		0.24					
[14]	10	Respiratory diseases	Respiratory disease score										0.70									0.6						
[19]	<75	NPC	NPC status	−0.32	−0.43	−0.35		−0.27					−0.4	−0.24														
[8]	N/A	Poor testicular function	Hb, Hct, TG, HDL-c, TC/HDL-c, uric acid	−0.23			−0.23						−0.21 ~ −0.47						0.31	0.28	0.27			0.37	0.27			
[20]	18–65	All-cause and cause-specific mortality	All cause death risk	NA	NA	NA		NA					NA										NA	NA	NA			
[17]	20–45	Anemia	Hb, Hct, RBC, WBC, CRP											0.3					0.2~ 0.5			0.4 ~ 0.5	0.2 ~ 0.3	0.2 ~ 0.3	0.2 ~ 0.3			
[16]	20–45	Abnormal liver function	ALT, AST, γ-GT, ALP, LDH, albumin, total bilirubin	−0.24	−0.3	−0.31								−0.3						0.47			0.46		0.22			
[13]	20–64	Hyperlipidemia & MetS	RBC aggregation, Hepcidin, %TS, sCD163	−0.25 ~ −0.29		−0.22						−0.21r ~ 0.38n	−0.3								0.28					−0.34		
[7]	20–64	Hypogonadism	Total testosterone, insulin, %TS, RBC aggregation	−0.25								−0.31n	0.26		0.35			0.24									−0.28	0.24
[21]	20–64	MetS	AST, RBC	−0.2 ~ −0.29		−0.24	−0.21					0.38	−0.24					0.25								−0.41	0.26	
[3]	20–65	Central obesity & MetS	Hepcidin, ferritin, ALT, HDL-c								0.23		−0.24								0.41	0.29 chicken & pork		0.22	0.34	−0.28		0.28
[5]	≧19	Hyperuricemia in women	Uric acid	−0.24 ~ −0.4		−0.24	−0.29				−0.29			−0.33			0.21											
[5]	≧19	Hyperuricemia in men	Uric acid	−0.2 ~ −0.33		−0.22	−0.31				−0.38						0.23					−0.24 lean meat	0.34	0.22	0.30			
[4]	≧19	Morbid obesity	BMI		−0.36			−0.16	−0.2 oat	−0.35	−0.16		−0.28									0.24 red meat	0.42		0.17			
[10]	≧40	Dyslipidemia in males; anemia in both gender	CRP, N/L ratio	−0.2	−0.4 ~ −0.5	−0.3			−0.3					0.2 ~ 0.3		0.2 ~ 0.3						0.2 ~ 0.3	0.2 ~ 0.3	0.2 ~ 0.3	0.2 ~ 0.3			
[11]	≧40	Weight status & others	WHR, TG, LDL-c, TC/HDL-c, BUN, creatinine	−0.23	−0.24		−0.22								−0.23				0.25 ~ 0.32	0.31	0.21	0.32		0.31	0.38			
[9]	≥40	MetS	WC, TG, HDL-c, SBP, DBP, FBG		−0.2							0.2 ~ 0.3		0.2 ~ 0.3	−0.2 ~ −0.3				0.2 ~ 0.3	0.3 ~ 0.4	0.2 ~ 0.3	0.3		0.3 ~ 0.4	0.4			
[18]	≧53	Frailty	Frailty score	−0.21	−0.4	−0.27 ~ −0.35		−0.46				−0.41		−0.23								−0.33						
[15]	≧65	Frailty	Frailty score	−0.33	−0.48	−0.2 ~ −0.23		−0.34	−0.27	−0.39			−0.21															
[6]	≧65	MCI in men	MMSE score		0.47	0.20 ~ 0.21			0.29oat ~ 0.33	0.46	0.24		0.24															
[6]	≧65	MCI in women	MMSE score	0.23	0.52		0.26	0.33	0.28	0.34	0.20			0.3														

%TS: serum transferrin saturation, ALP: alkaline phosphatase, ALT: alanine transaminase, AST: aspartate transaminase, BUN: blood urea nitrogen, CRP: C-reactive protein, DBP: diastolic blood pressure, FBG: fasting blood glucose, Hb: hemoglobin, HbA1C: glycated hemoglobin, Hct: hematocrit, HDL-C: high-density lipoprotein cholesterol, LDH: lactate dehydrogenase, LDL-C: low-density lipoprotein cholesterol, MCI: mild cognitive impairment, MetS: metabolic syndrome, MMSE: Mini-Mental State Examination, NA: not applicable, N/L: neutrophil-to-lymphocyte ratio, NPC: Nasopharyngeal carcinoma, RBC: red blood cells, SBP: systolic blood pressure, sCD163: soluble cluster of differentiation 163, TC/HDL-C: total cholesterol to high-density lipoprotein cholesterol ratio, TG: triglycerides, WBC: white blood cells, WC: waist circumference, WHR: waist-to-hip ratio, γ-GT: gamma-glutamyltransferase. ^a^ Factor loading is presented for each food item in each study. Green color denotes protective food items. Orange denotes harmful food items. Yellow denotes food items with mixed findings. When most of the references presenting in this table used reduced rank regression for data mining for dietary patterns, reference [19] applied partial least square discriminant analysis and reference [20] applied univariate food item selection method. ^b^ Respiratory diseases, including allergic rhinitis, current wheezing, and bronchitis; Weight status & others, including increased cardiovascular risk, and severity of impaired kidney function. ic rhinitis, current wheezing, and bronchitis; Weight status & others, including increased cardiovascular risk, and severity of impaired kidney function.

**Table 3 nutrients-14-02774-t003:** Taiwanese mean intake, percentile intake, and recommendation serving numbers for the 6 food groups ^a^.

	*N* = 6538					
Food Groups (Servings) ^b^	Mean	25th Percentile	Median	75th Percentile	Taiwanese Food Guide	Health TEA Recommendation
**Total carbohydrate-rich food**	**12.5**	**7.9**	**11.4**	**15.9**	**12.0**	**12.0**
Cereals and roots	7.8	4.2	7.0	10.3		>11
Carbohydrate-rich convenience foods	2.0	0.0	0.3	3.1		
Pastries and cookies	1.0	0.0	0.0	1.2		
Soup and miscellaneous foods	0.1	0.0	0.0	0.0		
Simple sugar	1.7	0.1	0.7	2.4		<1
**Total protein-rich food**	**7.7**	**4.0**	**6.5**	**10.1**	**5.0**	** 6.0 **
Soy bean and products	1.1	0.0	0.1	1.5		2.0
Fish and seafood	1.3	0.0	0.4	1.8		2.0
Eggs	0.8	0.0	0.5	1.1		1.0
Livestock and Poultry	4.1	1.2	3.1	5.7		1.0
Protein from staple foods	0.4	0.0	0.1	0.5		
**Dairy products**	**0.5**	**0.0**	**0.0**	**0.7**	**1.5**	** 0.5 **
**Vegetables**	**2.4**	**0.9**	**1.9**	**3.3**	**3.0**	**3.0**
**Fruits**	**1.5**	**0.0**	**1.0**	**2.2**	**2.0**	**2.0**
**Oil**	**5.3**	**2.4**	**4.3**	**7.1**	**5.0**	**5.0**
Cooking oil	4.8	2.1	3.9	6.5	4.0	4.0
Nuts	0.5	0.0	0.0	0.0	1.0	1.0
	***n* = 5549**					
**Sugar-free tea** (times/day)	**0.43**	**0**	**0.08**	**0.71**		1
	***n* = 5548**					
**Sugar-free coffee** (times/day)	**0.26**	**0**	**0**	**0.29**		

TEA: Taiwanese eating approach; ^a^ Data of 24 h recall from NAHSIT 2017–2020 (population age ≥19 years). Red color shows the basis of TEA recommendation. Underlined are recommendations made by this article. ^b^ Definition of serving size for six food groups: Total carbohydrate-rich food: One serving contains 15 g of carbohydrate; Total protein-rich food: One serving contains 7 g of protein; Dairy products: One serving contains 8 g of protein; Vegetables: One serving contains 25 kcal of calorie; Fruits: One serving contains 60 kcal of calorie; Oil: One serving contains 5 g of fat.

## Data Availability

Data of Nutrition and Health Survey from Taiwan described in the manuscript belong to HPA, which can be accessed in data center with the permission of HPA.

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
