# Peer review of "Healthy Taiwanese Eating Approach (TEA) toward Total Wellbeing and Healthy Longevity"

_nutrients, 2022, doi:10.3390/nu14132774_

Round 1
Reviewer 1 Report
It is an interesting article reviewing dietary patterns in Taiwanese. Based on the dietary patterns identified in Taiwanese, the authors discussed healthy and unhealthy foods, then compared food amounts from the national nutrition surveys with the national dietary guidelines and thus provided additional recommendations on some food intakes.
Some comments for authors to consider for revision are as follows.
- The searching strategies and the type of this review were not clear enough addressed, although the authors mentioned ‘include only dietary factors highly associated…to find linear combination…’, the 1st paragraph, Materials and methods. Were any dietary patterns published by other statistical analyses except for the reduced rank regression and partial least square discriminant analysis, if any, such as the principal component method of factor analysis and latent variable analysis? Were there any dietary pattern studies archives in other scientific databases such as Cochrane? It would be better if the authors could provide a data search flow chart or explain it in the Method.
- Titles of tables should be given, and all abbreviations should be indicated.
- There were 15 cross-sectional studies of the 17 manuscripts with multiple health outcomes from several primary databases. Any limitations of this review should be mentioned in the Discussion.
- The food ‘roots’ had less information and was only from the national nutrition survey, not from the reviewed dietary pattern. Might it not be mentioned in the Abstract?
Author Response
Responses to reviewer 1
It is an interesting article reviewing dietary patterns in Taiwanese. Based on the dietary patterns identified in Taiwanese, the authors discussed healthy and unhealthy foods, then compared food amounts from the national nutrition surveys with the national dietary guidelines and thus provided additional recommendations on some food intakes.
Some comments for authors to consider for revision are as follows.
- The searching strategies and the type of this review were not clear enough addressed, although the authors mentioned ‘include only dietary factors highly associated…to find linear combination…’, the 1st paragraph, Materials and Were any dietary patterns published by other statistical analyses except for the reduced rank regression and partial least square discriminant analysis, if any, such as the principal component method of factor analysis and latent variable analysis? Were there any dietary pattern studies archives in other scientific databases such as Cochrane? It would be better if the authors could provide a data search flow chart or explain it in the Method.
Ans:
Thank you for your suggestion. Now a data search flow chart is included. Please see figure 1. Accordingly, we rewrote the section on“the Strategy to search Taiwanese dietary pattern related literature”.
- Titles of tables should be given, and all abbreviations should be
Ans:
Titles of tables and all abbreviations are provided and all abbreviations are indicated.
- There were 15 cross-sectional studies of the 17 manuscripts with multiple health outcomes from several primary Any limitations of this review should be mentioned in the Discussion.
Ans:
Thank you very much for your suggestion. We added a paragraph on the limitations of this review.
Reviewer 2 Report
Thank you for the opportunity to review this manuscript. The authors have reviewed dietary patterns and their potential impact on health in Taiwan. There are several concerns for the authors to consider.
My major concerns:
1. The manuscript is entitled “Healthy Taiwanese Eating Approach (TEA) toward total wellbeing and healthy longevity”. However, I have not seen how “healthy longevity” was defined in the manuscript. “Healthy longevity” is not mentioned in the whole text except the title. How dietary patterns are associated with healthy longevity is not clear.
2. The strategy of the literature search is not well described: how many records were identified via initial searching? How many articles were excluded from the review? How many articles were included in the review? It is better to provide a flowchart to display the literature search.
3. The authors have included many results in the Methods section. Please consider moving these results to Results section.
4. Limitations or strengths have not been mentioned in the study.
5. To my understanding, the authors have firstly reviewed previous publications and then analysed dietary intakes using the Nutrition and Health Survey from Taiwan. I cannot find any new knowledge provided by the review.
6. Most of the studies were conducted among adults and only several studies investigated dietary patterns in children. The findings from these several studies among children seem not to provide valuable information. Using the Nutrition and Health Survey from Taiwan, the authors only analysed dietary intakes among adults but not children. Why not just include those studies among adults in the review?
Minor concerns:
1. Please note that line and page numbering would facilitate the review process.
2. There are many abnormal marks in Table 2.
3. No titles for Tables are included. Please make revision.
4. What are the numbers in Table 2?
Author Response
Responses to reviewer 2
Thank you for the opportunity to review this manuscript. The
authors have reviewed dietary patterns and their potential
impact on health in Taiwan. There are several concerns for the
authors to consider.
My major concerns:
- The manuscript is entitled “Healthy Taiwanese Eating
Approach (TEA) toward total wellbeing and healthy longevity”.
However, I have not seen how “healthy longevity” was defined in
the manuscript. “Healthy longevity” is not mentioned in the whole
text except the title. How dietary patterns are associated with
healthy longevity is not clear.
Ans:
Thank you for your comments. Several articles are pertaining to frailty and cognitive impairment which are indicators of healthy longevity. There is one article on the inverse association between healthy Taiwanese Eating approach and all-cause mortality and CVD mortality, which was discussed in the discussion. Now I included it in the result. You are absolutely right that we should make link to longevity throughout the text.
- The strategy of the literature search is not well described: how
many records were identified via initial searching? How many
articles were excluded from the review? How many articles were
included in the review? It is better to provide a flowchart to
display the literature search.
Ans:
Thank you. We have made the change accordingly. Please see Figure 1 for the flow-chart.
- The authors have included many results in the Methods
section. Please consider moving these results to Results
section.
Ans:
Accordingly, we now moved the description of the discovered studies into results.
- Limitations or strengths have not been mentioned in the study.
Ans:
In the discussion, we now added a section on limitations.
- To my understanding, the authors have firstly reviewed
previous publications and then analysed dietary intakes using
the Nutrition and Health Survey from Taiwan. I cannot find any
new knowledge provided by the review.
Ans:
In the past, most dietary pattern studies are published with respect to one specific health condition or disease. Most of the healthy dietary patterns reported include abundant well-known protective dietary food groups such as vegetables, fruits, whole grains, but less processed foods. Although we repeatedly see similarities across countries and across outcomes; very rarely, there are studies integrating multiple health outcomes in one presentation to illustrate the point that a prudent dietary pattern is pertaining to protection of all diseases. On the other hand, authors in past studies did not compare and show some subtle differences for different health outcomes. This was what we have done. Unlike medicines, specific single compound is often used for each symptom; a prudent diet works for preventing most diseases and ensuring healthy longevity. That is the point we would like to stress.
As to why we also presented side by side the current Taiwanese dietary profile, particularly 25th, 50th, and 75th percentiles of each food groups; this information is there for readers to understand the meaning of the higher intake or lower intake levels in this population. More or less of certain foods may mean differently in different populations.
- Most of the studies were conducted among adults and only
several studies investigated dietary patterns in children. The
findings from these several studies among children seem not to
provide valuable information. Using the Nutrition and Health
Survey from Taiwan, the authors only analysed dietary intakes among adults but not children. Why not just include those studies among adults in the review?
Ans:
The purpose of including studies on children, adults, and elderlies is to see whether there is consistency among dietary patterns for disease at young age, mid-life, and older age. Indeed we found consistency of protective or harmful foods irrespective of when the health outcomes take places with minor inconsistency. Then we provided current Taiwanese profile, showed its distance with Taiwanese food guide, and make further suggestions. If we would like to make recommendation specific to children, we need more studies to tackle multiple aspect of childhood diseases in detail. That is not our purpose.
Minor concerns:
- Please note that line and page numbering would facilitate the
review process.
Ans:
Thank you for the reminder.
- There are many abnormal marks in Table 2.
Ans:
Thank you. These have been taken care of.
- No titles for Tables are included. Please make revision.
Ans:
Thank you. Absolutely.
- What are the numbers in Table 2?
Ans:
They are counts for protective and harmful findings. We have now added footnotes to clarify them.

Round 2
Reviewer 2 Report
Thank the authors for taking my comments into consideration. The authors have much improved their manuscript, but there are still some concerns for the authors to consider:
1. The literature search terms are not sufficient. For example, “diet quality” was not included in the search.
2. Why not include the publications investigating the diet patterns defined by diet index?
3. Why there are different numbers of included studies in the present version (n=19) and the previous version (n=17)?
4. Table 1: there are not association figures for reference 20.
Author Response
Thank the authors for taking my comments into consideration. The authors have much improved their manuscript, but there are still some concerns for the authors to consider:
- The literature search terms are not sufficient. For example, “diet quality” was not included in the search.
Ans:
Thank you for your comments. As shown in our searching strategy flow chart, we excluded articles using the pre-determined dietary patterns including Mediterranean diet, dash diet, healthy diet, western diet. The purpose of our study was to gather patterns completely out of data mining and to summarize them together. "Diet quality' estimation equations are mostly derived from dietary guideline and do not fit into our study purpose.
- Why not include the publications investigating the diet patterns defined by diet index?
Ans:
The answer is the same as that of question 1.
- Why there are different numbers of included studies in the present version (n=19) and the previous version (n=17)?
Ans:
We used "Taiwan*" instead of "Taiwan" in the search for our revised manuscript. In this search, two more papers were found due to the key word of "Taiwanese".
- Table 1: there are not association figures for reference 20.
Ans:
We have explained this in the manuscript in both in the paragraph of “Data summary and presentation method” in Materials and methods, and also in the footnote under Table 1. Below are the extracted sentences.
“No loading coefficients provided for one article (20) which find significantly associated food items by univariate logistic regression method”
“When most of the references presenting in this table used reduced rank regression for data mining for dietary patterns, reference 19 applied partial least square discriminant analysis and reference 20 applied univariate food item selection method.”